# Adolescent resilience and mobile phone addiction in Henan Province of China: Impacts of chain mediating, coping style

Anna Ma[1¤], Yan Yang[1], Shuangxi Guo[2], Xue Li[1], Shenhua Zhang[3], Hongjuan Chang[1]*

**1** School of Nursing, Xinxiang Medical University, Xinxiang, Henan Province, China, **2** Department of Neurology, The First Affiliated Hospital of Xinxiang Medical University, Xinxiang, Henan province, China, **3** Weihui Senior Middle School, Xinxiang, China

¤ Current address: School of Nursing, St. Paul University, Manila, Philippines
* changhj0812@126.com

## Abstract

### Background

As mobile phone use grows, so it brings benefits and risks. As an important part of adolescents healthy growth, resilience plays an indispensable role. Thus, it is important to identify when mobile phone use of an adolescent becomes an addiction. This study proposed to explore the effects of adolescent resilience on mobile phone addiction, and tested the mediating role of coping style and depression, anxiety, and stress (DASS) on phone addiction among 2,268 adolescents in the Henan province, China.

### Methods

The adolescents were surveyed via an online questionnaire, a mobile phone addiction index (MPAI), a depression, anxiety, and stress scale with 21 items (DASS-21), the Resilience Scale for Chinese Adolescents (RSCA), and the Simplified coping style questionnaire (SCSQ), and we used structural equation modeling to examine the correlations and moderation effects. All data analyses were performed using SPSS 26.0 and Amos 23.0.

### Results

The results show that adolescences resilience were negatively related to negative coping, DASS, and mobile phone addiction; both coping style and DASS could mediate the relationship between adolescent resilience and mobile phone addiction among Chinese adolescents. The relationship between adolescent resilience and mobile phone addiction in Chinese adolescents was mediated by the chain of coping styles and DASS.

### Conclusions

There is a negative relationship which exists between resilience and mobile phone addiction in this population. In addition, stress, anxiety, depression, and coping style significantly

**Data Availability Statement:** All relevant data are within the paper and its Supporting information files.

**Funding:** This work was supported by the National Natural Science Foundation of China (grant number: 81803252). This funder had no role in the design of the study and collection, analysis, and interpretation of data and in preparation and writing the manuscript.

**Competing interests:** The authors have declared that no competing interests exist.

**Abbreviations:** DASS, depression, anxiety, and stress; MPAI, mobile phone addiction index; RSCA, the Resilience Scale for Chinese Adolescents; SCSQ, the Simplified coping style questionnaire.

influence the risk of adolescent mobile phone addiction and play an intermediary role in Chinese adolescent resilience and mobile phone addiction.

## Introduction

With the development of information technology, the functions of a mobile phone are becoming more and more powerful; it is now considered as an important communication tool and social accessory. However, increased dependency brings a downside, namely, smartphone phone addiction. Smartphone addiction consists of four main components: obsessive phone use and compulsive behaviors; tolerance of longer and more intense of use; withdrawal or suffering without the phone; and functional impairment [1].

Due to the COVID-19 outbreak, mobile phones have become increasingly important for online teaching and learning in China. According to the 48th China statistical report on internet development, by June 2021, there were 1.07 billion mobile phone users in China, accounting for 99.7% of the total number of internet users. Furthermore, internet users spend an average of 26.9 hours online per week, and the number of internet users between the ages of 6 and 19 reached 158 million, accounting for 15.7% of the total [2]. As part of normal adolescent psychological development, this age group develops susceptibility to peer influences and tends to have low-risk perception—factors that can result in increased risk-taking behavior and poor self-regulation [3]. Stressful life events have been identified as a significant risk factor in the emotional state of adolescents. The spread of COVID-19 has been one of these stressful life events as it led to substantial social and economic changes as a result of many governments introducing quarantine policies to prevent the virus from spreading and thus assure the safety of the population. Psychopathologists have focused on the psychological impact of COVID-19 and its variations in the adolescent population and they have identified moderate to severe levels of stress, anxiety, and depression in this population [4]. Guang-Li found that a moderate positive correlation between negative coping style and adolescents' mobile phone addiction [5].

In the middle of July 2021, Henan province suffered from unusually heavy rainfall (continuous rainfall of 958 mm [6]), causing severe flooding. The flood, named the "7.20 Henan rainstorm," overwhelmed dams and burst riverbanks in a short time, causing severe traffic paralysis, power failure, loss of crops, and upending tens of millions of lives. Randeniya reported that after this natural disaster, adolescents of Sri Lanka were mostly affected by sleeping difficulties [7]. Makwana indicated that the psychological effects of the disaster were fiercer among children, women, and the dependent elderly population [8].

The general strain theory argues that adolescents can be pressured into delinquency by a negative experience, this negative effect creates pressure for corrective action and may lead adolescents to escape from the source of their adversity, or manage the negative effect through the use of illicit drugs [9]. The general strain theory was originally used for adolescents' criminal behavior, but now many researchers use it for explaining addictive behaviors [10–12]. Recent studies have argued that stress and disaster can cause mobile phone addiction through various pathways [13–15]. Disaster can lead to severe stress, uncontrollable stress, and substance dependency of the individual [8]. According to many studies, natural disasters and COVID-19 as stressors can lead people to depression, anxiety, substance abuse, and addictive behavior [16, 17]. Some experts also believe that depression, anxiety, and pressure may lead to

internet addiction [18]. Research has also shown that exposure to disaster is positively associated with substance abuse and is negatively related to children's psychological resilience [19].

Resilience is commonly described as the ability to bounce back or overcome some form of adverse event or situation and thus experience positive outcomes [20]. Resilience is an important developmental stage during adolescence, as it is a transitional period characterized by significant neurobiological and psychosocial changes in the context of amplifying environmental demands and increasing sensitivity to social contexts [21]. One study has found that there is a negative relation between high psychological resilience and depression [22]. Resilience is also a protective factor of excessive smartphone use [23]. Many scholars believe that adolescent resilience is also related to substance abuse, such as smoking and excessive drinking (alcoholism) [24]. However, studies on the relationship between mental resilience and adolescents' coping styles, mental health, and mobile phone addiction are rare.

In view of this, adolescents confined to their homes due to the floods in Henan may have overused their mobile phones and the internet. Many studies have shown that mobile phone addiction has had a negative impact on physical and mental health as well as social adaption, academic achievement, and sleep quality [25–27]. Thus, we hypothesized that there is a correlation between resilience and mobile phone addiction among adolescents and that coping style and mental health play a mediating role in that relationship.

"Coping" means constantly changing cognitive and behavioral efforts to manage specific external and/or internal demands that are appraised as taxing or exceeding a person's resources [28]. Chapman thought that adolescents with lower self-esteem engage in coping strategies of ventilating feelings, avoiding problems, and relaxing, while adolescents with higher self-esteem were more likely to engage in coping styles that directly address solving the problem [29]. During the COVID-19 pandemic, children who used positive strategies to cope with the situation suffered less emotionally and behaviorally [30]. Fang Liu found that a negative coping style mediated the relationship between smartphone addiction and childhood psychological maltreatment [31]. Psychological resilience is the ability to cope with a crisis or to quickly return to a pre-crisis status, either mentally or emotionally [32].

Resilience negatively predicted a negative coping style and positively predicted a positive coping style. Life events not only directly influenced a negative coping style and positive coping style but also indirectly influenced coping styles by affecting resilience [33]. The mutually enhancing relationship between resilience and positive mental health, and vice versa, a mutually reducing relationship between resilience and mental illness, and presented the significant influence of mental health level on resilience [34]. Malek showed that the coping style of avoidance could aggravate depressive, anxiety, and stress symptoms of depression, anxiety, and stress in participants during the COVID-19 pandemic. Keeping optimism, resilience, and approach coping styles decreased the mental health burden of the pandemic on participants [35].

## The present study

Based on the literature review above, the relationship between adolescent resilience and mobile phone addiction deserves attention. The present study constructed a chain mediation model to examine the mediating role of negative coping, stress, anxiety, and depression in the relationship between adolescent resilience and mobile phone addiction among Chinese adolescents. Furthermore, we proposed a model to test the associations among Chinese adolescent resilience, coping style, mental health, and mobile phone addiction, to further clarify mobile phone addiction related to resilience Fig 1.

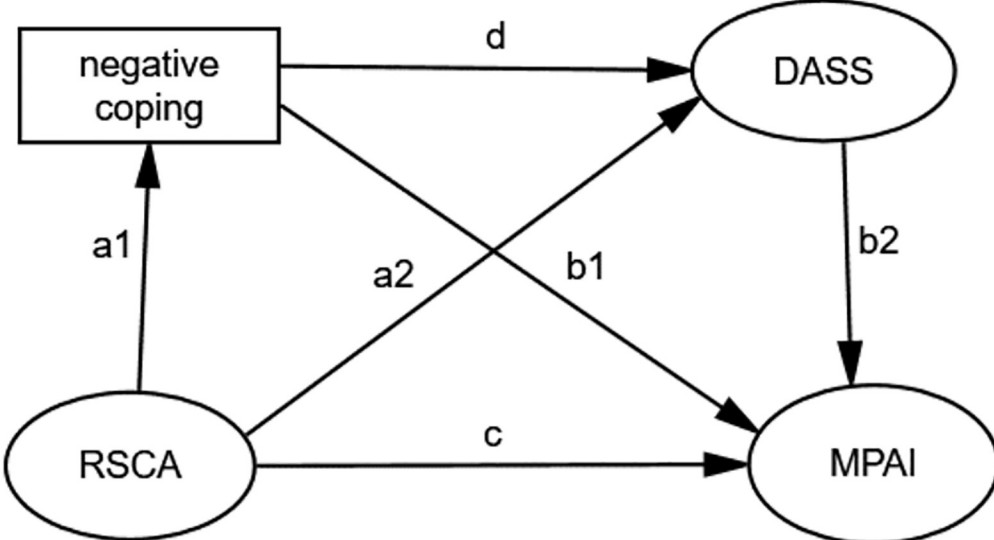

**Fig 1. Hypothesized model.** Notes: mobile phone addiction index (MPAI); depression, anxiety, and stress scale with 21 items (DASS-21); Resilience Scale for Chinese Adolescents (RSCA); Simplified coping style questionnaire (SCSQ).

## Materials and methods

### Participants

The convenience sampling method was employed to select adolescents from middle school and high school in Henan province of China to complete the questionnaires online from July 1 to August30, 2021. In total, 2,268 valid questionnaires were obtained, with an effective rate of 97.28%. Among them, the mean age was 14.90 years ($SD$ = 2.58, range = 12–21 years), participants including 979 boys(43.20%)and 1,289 girls (56.80%). They completed a survey that included demographic variables, a mobile phone addiction index (MPAI), a depression, anxiety, and stress scale with 21 items (DASS-21), the Resilience Scale for Chinese Adolescents (RSCA), and the Simplified Coping Style questionnaire (SCSQ). The link to the questionnaires was sent out to potential participants. All questions were completed by respondents clicking on the appropriate option. In the case of a missing answer, the server would remind them to give the missing answer. All participants filled in the questionnaire anonymously, without any payment. Their privacy was protected.

### Ethics approval and consent to participate

The participants were under 18 years old, and we provided written informed consent to their parents (or legal guardians)before taking part in the study. The procedures performed in the study were approved by the Ethics Committee of Xinxiang Medical University(number:XYLL-2018015), as are all studies involving human participants. The data from the participants were de-identified.

### Measurement of structures

**DASS-21.** The DASS-21 was used to evaluate negative emotional states of depression, anxiety, and stress [36] and refer to the previous week. In the DASS-21, each item is classified into four Likert responses from 0 to 3, where 0 = "nothing" and 3 = "Most of the time." This self-report instrument includes three subscales: 1) the stress subscale, which measures tension,

agitation, difficulty relaxing, and negative affection; 2)the anxiety subscale, which assesses autonomic arousal, skeletal musculature effects, situational anxiety, and subjective experience of anxiety arousal; and 3) the depression subscale, which measures hopelessness, dysphoria, lack of interest, self-deprecation, and inertia. The reliability coefficients of depression, anxiety and stress were 0.82, 0.82 and 0.79, respectively. The Cronbach's alpha of the total scale was 0.89.

**MPAI.** The MPAI was designed by Louis Leung to identify addiction symptoms associated with mobile phone use among adolescents in Hong Kong [37]. The scale includes 17 items answered on a five-point Likert scale of 1 to 5 (1 = not at all; 2 = rarely; 3 = occasionally; 4 = often; and 5 = always). The scale covers four dimensions: 1) "inability to control craving," which reflects the amount of time an adolescent spends on the mobile phone, thereby leading to complaints from family and friends about their compulsive mobile phone use and causing the adolescent's loss of sleep due to the excessive use; 2) "Anxiety and feeling lost" assesses preoccupation, feeling lost or anxious, and having difficulty switching off the mobile phone; 3)"Productivity loss" measures decreased productivity and diverted attention from pressing issues due to an adolescent's excessive use mobile phones; 4) "Withdrawal and escape" indicates that an adolescent uses their mobile phones to escape from isolation, loneliness, and feeling down. The Cronbach's alpha of scale was 0.90.

**RSCA.** The RSCA was developed by Yueqin Hu [38]according to the process model of the resilience concept and applied to Chinese adolescents. There are 27 items divided into two factors: "manpower" and "support". The former includes three factors: goal focus, emotion control, and positive cognition while latter includes two factors: family support and interpersonal assistance. The reliability of the total scale was 0.85.

**SCSQ.** This SCSQ was designed by Ya-Ning Xie [39], composed of 20 items, Coping styles can be divided into two groups based on how people react to setbacks: "positive coping" and "negative coping." The reliability of the total scale was 0.90, while the positive coping and negative response subscales were 0.89 and 0.78, respectively.

### Data analysis

All data analyses were performed using SPSS 26.0 and Amos 23 (IBM Inc., Armonk, NY, USA). First, descriptive data were received using SPSS 26.0, and correlations variables were calculated using Pearson's correlations. Second, according to Baron and Kenny [40], we analyzed the mediation effects using two measurement models to examine how well the indicators represented each latent variable. Second, we tested the hypothesized relationships among latent variables. Maximum likelihood (ML) estimation was used to test the two structural models in the AMOS 23.0 program. When TLI > 0.90, CFI > 0.90, and RMSEA < 0.06, the model fits well, according to Hu and Bentler [41]. We followed the stepwise method to structure the best-fitting model for the mediated effects and bootstrapping with 5,000 replications to measure the chain mediation model. All data analyses were two-tailed, with significance levels of $P < 0.01$ and $P < 0.05$.

## Results

### Descriptive statistics

We included 2,268 participants, including 979 boys (43.20%) and 1,289 girls (56.80%), in the final analysis. The proportion of girls was slightly higher than that of boys (56.80% vs. 43.20%). The mean age was 14.90 years (SD = 2.58, range 12–21 years). There were 1244 (54.85%) participants from middle school, 1,024 (45.15%) from high school, 169 (7.50%) from one-child families, and 2,099 (92.50%) from multi-child families. The other results are shown in Table 1.

**Table 1. Demographic profiles and descriptive statistics of the participants.**

| Gender | Frequency | Percentage |
|---|---|---|
| boy | 979 | 43.2 |
| girl | 1289 | 56.8 |
| **One-child** | | |
| yes | 169 | 7.5 |
| no | 2099 | 92.5 |
| **Birth order** | | |
| 1st | 1048 | 46.2 |
| 2nd | 1041 | 45.9 |
| 3rd | 179 | 7.9 |
| **Nationality** | | |
| Han | 2259 | 99.6 |
| Hui | 8 | 0.4 |
| Miao | 1 | 0.0 |
| **Grade** | | |
| middle school 7th | 174 | 7.7 |
| middle school 8th | 634 | 28.0 |
| middle school 9th | 436 | 19.2 |
| high school 1st | 34 | 1.5 |
| high school 2nd | 471 | 20.8 |
| high school 3rd | 519 | 22.9 |
| Total | 2268 | 100.0 |

## Univariate analysis

The results for the 2,268 participants are displayed in Table 2. The category totals as total mean (*SD*) were as follows: MPAI, 39.57 (±13.82); DASS-21, 5.190 (±4.57); positive coping was 22.45 (±9.18); negative coping, 11.93 (±5.64); RSCA, 88.89 (±18.50).

## Correlation analysis of major study variables

The variables correlated with the constructs in Table 3 were less than 0.85. The discriminant validity value (< 0.85) was met in the construct correlation [42]. These findings showed that valid and reliable constructs were used. We found that adolescents resilience were negatively related to negative coping, DASS, and mobile phone addiction. Negative coping was positively related to both DASS and mobile phone addiction. The results of analyses also showed that adolescent resilience correlated with negative coping (r = -.317, $P < 0.01$), DASS(r = -.593, $P < 0.01$), and mobile phone addiction(r = -.405, $P < 0.01$); negative coping correlated with DASS(r = .603, $P < 0.01$), mobile phone addiction(r = .322, $P < 0.01$); DASS correlated with mobile phone addiction(r = .448, $P < 0.01$). Adolescents with poor resilience usually adopted a negative coping style and had higher anxiety, depression and mobile phone addiction—which confirms the hypothesis.

## Structural model testing and structural relationship between constructs

The test results revealed the goodness of fit of the proposed structural model ($\chi 2/df$ = 2.57, RMSEA = 0.054, GFI = 0.978, CFI = 0.984). The hypothesis relationships between the variates are demonstrated in Table 4. Results of the test for path(negative coping <- - -RSCA) revealed a significant negative relationship between resilience and negative coping ($\beta$ = -1.025,

**Table 2. Basic characteristics and measure scores.**

|  | M | Std.Error of Mean | Frequency | Percentage |
|---|---|---|---|---|
| **Age** | 14.90 | 2.58 | | |
| **MPAI Total** | 39.57 | 13.82 | | |
| feeling anxious&lost | 7.96 | 4.02 | | |
| inability to control craving | 16.65 | 5.92 | | |
| productivity loss | 7.64 | 3.28 | | |
| withdrawal | 7.32 | 3.49 | | |
| **DASS-21 Total** | 5.19 | 4.57 | | |
| **Stress** | 7.40 | 5.07 | | |
| normal | | | 2054.00 | 90.56 |
| mild | | | 124.00 | 5.47 |
| moderate | | | 76.00 | 3.35 |
| severe | | | 14.00 | 0.62 |
| extremely severe | | | 0.00 | 0.00 |
| **Anxiety** | 6.31 | 4.94 | | |
| normal | | | 1583.00 | 69.80 |
| mild | | | 218.00 | 9.61 |
| moderate | | | 304.00 | 13.40 |
| severe | | | 103.00 | 4.54 |
| extremely severe | | | 60.00 | 2.65 |
| **Depression** | 6.58 | 5.09 | | |
| normal | | | 1773.00 | 78.18 |
| mild | | | 258.00 | 11.38 |
| moderate | | | 179.00 | 7.89 |
| severe | | | 48.00 | 2.12 |
| extremely severe | | | 10.00 | 0.44 |
| **SCSQ Total** | 56.39 | 17.91 | | |
| positive coping | 22.45 | 9.18 | | |
| negative coping | 11.93 | 5.64 | | |
| **RSCA Total** | 88.89 | 18.50 | | |
| focuced | 16.24 | 4.88 | | |
| Interpersonal support | 18.51 | 5.90 | | |
| emotional control | 19.63 | 5.76 | | |
| Positive cognitive | 14.30 | 3.78 | | |
| familiy support | 16.77 | 4.50 | | |

**Table 3. Correlation analysis of study variables.**

|  | 1 | 2 | 3 | 4 |
|---|---|---|---|---|
| **1.RSCA Total** | 1 | | | |
| **2.Negative coping** | -.317** | 1 | | |
| **3.DASS-21 Total** | -.593** | .603** | 1 | |
| **4.MPAI Total** | -.405** | .322** | .448** | 1 |

Note:

$^{**}P < 0.01$

**Table 4. Results of the structural model: Tests of hypothesized associations between constructs.**

|  |  |  | Estimate | S.E. | t-value | P |
|---|---|---|---|---|---|---|
| negative coping | <--- | RSCA | -1.025 | 0.078 | -13.216 | *** |
| DASS | <--- | negative coping | 0.292 | 0.017 | 16.871 | *** |
| DASS | <--- | RSCA | -1.152 | 0.061 | -18.992 | *** |
| MPAI | <--- | RSCA | -0.836 | 0.07 | -11.919 | *** |
| MPAI | <--- | negative coping | 0.09 | 0.017 | 5.355 | *** |
| MPAI | <--- | DASS | -0.096 | 0.031 | -3.092 | 0.002 |

Note:

***P < 0.01

t = -13.216, $P < 0.01$). Results of the test for path(DASS <---negative coping) showed a significant positive relationship between DASS and negative coping ($\beta = 0.292$, t = 16.871, $P < 0.01$). Results of the test for path(DASS <---RSCA) revealed a significant negative relationship between DASS and resilience ($\beta = -1.152$, t = -18.992, $P < 0.01$). Results of the test for path(MPAI <---RSCA)showed a significant negative association between mobile phone addiction and resilience ($\beta = -0.836$, t = -11.919, $P < 0.01$). Results of path(MPAI <---negative coping)showed a significant relationship positive relationship between mobile phone addiction and negative coping ($\beta = 0.09$, t = 5.355, $P < 0.01$). Results of the test for path(MPAI <---DASS)revealed a significant negative relationship between mobile phone addiction and DASS ($\beta = -0.096$, t = -3.092, $P = 0.002$). Finally, the results confirmed the hypothesis. The variables were explained by their respective preceding construct (Fig 2). The indirect effects are presented in Table 5. Bootstrapping (the process was repeated 5,000 times)analyses showed that the indirect effects of adolescent resilience on mobile phone addiction through negative coping and stress, anxiety, and depression were significant and positive (standardized indirect effect 0.029, 95%CI [0.012,0.048], $P < 0.01$), and the indirect effect of adolescent resilience on mobile phone addiction through stress, anxiety, depression was 0.111, 95% CI [0.045, 0.186], $P < 0.01$, excluding 0, and the mediating effect was significant. The indirect effect of adolescent resilience on mobile phone addiction through negative coping was -0.092, 95% CI [-0.125, -0.061], $P < 0.01$, excluding 0, and the mediating effect was significant.

## Discussion

This study surveyed the ways by which adolescent psychological resilience, coping style and depression, anxiety, and stress affected mobile phone addiction among Chinese adolescents. The main aim of the study was to establish if there was a relationship between resilience and mobile phone addiction in Chinese adolescents and if negative coping and DASS were a chain mediator of this relationship. The results showed that adolescent psychological resilience could directly and negatively affect mobile phone addiction in Chinese adolescents, in consequence, poor resilience may cause mobile phone addiction in Chinese adolescents, which is consistent with previous research findings.

According to the psychological resilience framework theory [43], psychological resilience is an important protective factor for problem behavior and personal mental health. Griffiths argued that addictions consist of several components, such as relapse, mood modification, tolerance, conflict, and withdrawal [44]. In the studies, psychological resilience was found to protect the addictive behaviors (internet problematic use) [45, 46]. Adolescent mobile phone addiction affect their life and study, and this study suggests that family, peer, teacher support,

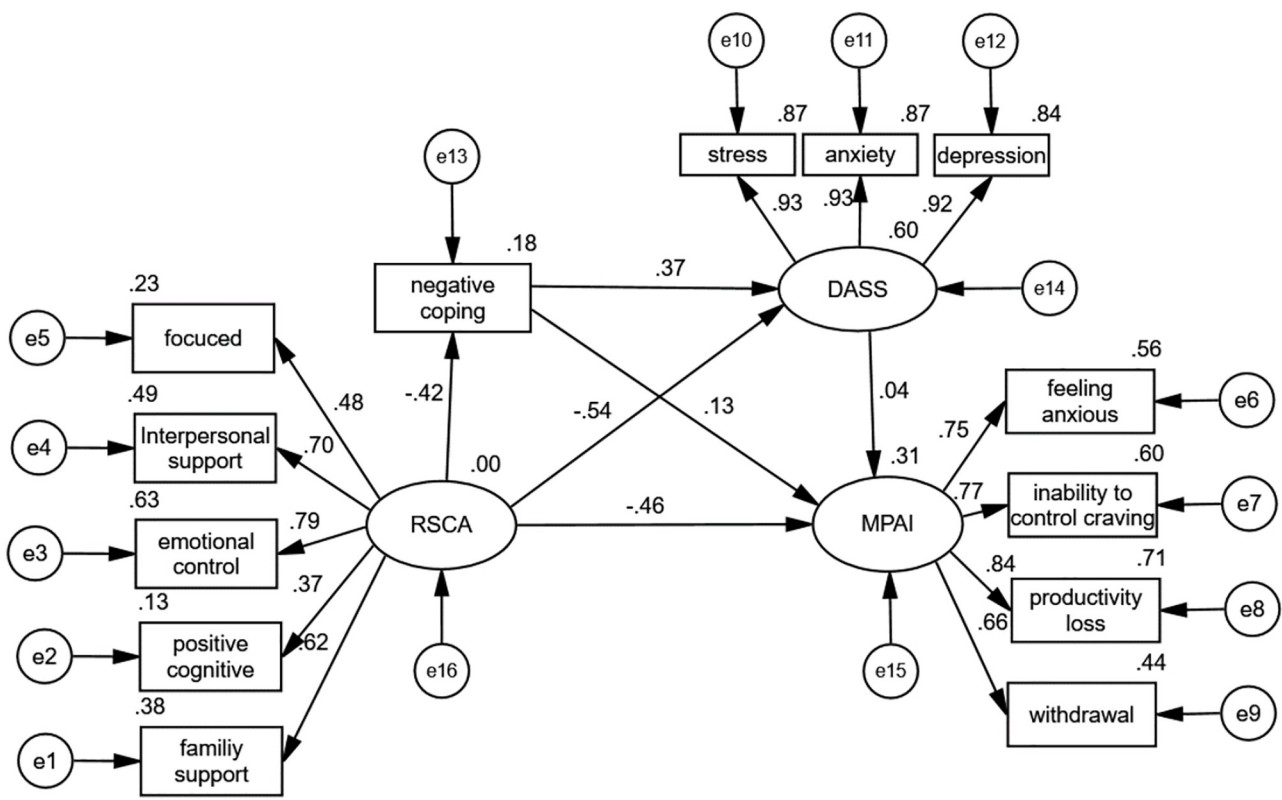

**Fig 2. The standardized path coefficients in model testing.**

and exercise help to improve the depressive moods and make it more resilient to adversity and stress.

The experimental results reveal that adolescents psychological resilience is negatively correlated with negative coping style: negative coping style is negatively correlated with mobile phone addiction. It shows that adolescents with good psychological resilience are more likely to be adopt positive coping styles when facing pressure and frustration and are less likely to addicted to mobile phones. Francisco found that higher scores on perceived stress represented lower resilience and lower use of the coping strategies of problem-focusing [47]. Li et al. [33] also found that resilience could negatively predict the negative coping style and positively predict the positive coping style, which is consistent with the results of this study. He et al. [48]

**Table 5. Bootstrap truncated regression results.**

| Relationships | Product of coefficients | | | Bootstrapping | | | | P |
|---|---|---|---|---|---|---|---|---|
| | | | | BC 95%CI | | Percentile 95%CI | | |
| | point estimate | SE | Z | Lower | Upper | Lower | Upper | |
| Indirect Effects | | | | | | | | |
| RSCA → negative coping → MPAI | -0.092 | 0.018 | -5.111 | -0.125 | -0.061 | -0.125 | -0.061 | 0 |
| RSCA → DASS → MPAI | 0.111 | 0.043 | 2.581 | 0.043 | 0.188 | 0.045 | 0.186 | 0.002 |
| RSCA → negative coping → DASS→ MPAI | 0.029 | 0.011 | 2.636 | 0.012 | 0.048 | 0.012 | 0.048 | 0.002 |
| Total | -0.789 | 0.052 | -15.173 | -0.889 | -0.698 | -0.89 | -0.699 | 0 |

found that adolescents adopt a negative coping style, which leads to mobile phone addiction. This study reveals that depression, anxiety, and stress (DASS)is negatively correlated with mobile phone addiction. Gao et al found that perceived stress was directly associated with mobile phone addiction [14]. A large number of studies have shown that mobile phone addiction is positively associated with anxiety and depression, which in turn brings about severe adverse effects on mental health, such as anxiety, depression and poor sleep quality [49–52]. These studies suggest an interplay between anxiety, depression, and phone addiction.

These results show that negative coping styles and depression, anxiety, and stress (DASS) play intermediary roles, respectively, in adolescent resilience and mobile phone addiction in Chinese adolescents; thus, our hypothesis was verified. This finding is consistent with the results of previous studies. For example, Stanković found partial mediation between internet addiction and depression through stress and anxiety [53].

The Simplified Coping Style Questionnaire(SCSQ) results revealed that coping style had a maximal effect on adolescent mobile phone addiction [5]. Many studies have indicated a relationship between depression, anxiety, and loneliness with smartphone usage [54, 55]. Depression and social anxiety are risk determinants for greater problematic smartphone use [56]. Stress, anxiety, and depression were significantly positively correlated with smartphone addiction [57]. Researcher have found that a significant positive relationship between anxiety about COVID-19 infection and daily smartphone use hours; the largest predictor of smartphone addiction was anxiety about COVID-19 infection [58].

Negative coping style and DASS played a continuous intermediary role in the impact of adolescent resilience on mobile phone addiction among Chinese adolescents. Smartphone users who experience depressive symptoms may similarly use their mobile devices as a coping strategy to alleviate these unpleasant symptoms [59]. Coping and affective disorders appear to play a key role in international addiction among adolescents [60]. COVID-19 and floods as stressors can cause psychological stress responses in adolescents, and differences in coping styles can cause differences in behaviors in adolescents. Coping style is a significant factor leading to smartphone addiction among adolescents. Problem-focused coping strategies indicate that coping behaviors directly target the source of stress and can prompt participants to use positive coping styles to deal with the adverse consequences of the pandemic; conversely, coping styles of avoidance, denial, and fantasy in dealing with stress make it a potentially strong risk factor for smartphone addiction [61, 62].

## Limitations

This study had several limitations. The convenience sample limits the universality of the results. Factors such as family environment, personality traits, peer relationships, and sleep quality may also affect mobile phone addiction among adolescents. Therefore, future studies should examine whether the relationship between Chinese adolescent resilience, coping style, DASS, and mobile phone addiction will change over time.

## Conclusions

This study explored the impact mechanism of the effect of resilience on mobile phone addiction among Chinese adolescents. The main aim of the study was to establish if there was a relationship between resilience and mobile phone addiction in Chinese adolescents and if negative coping and DASS were a chain mediator of this relationship. The structural equation model was utilized to synchronously examine the individual and continuous mediating roles of coping styles and DASS. The results of indicate that a negative relationship exists between resilience and mobile phone addiction in this population. In addition, stress, anxiety, depression,

and coping style significantly influence the risk of adolescent mobile phone addiction and play an intermediary role in Chinese adolescent resilience and mobile phone addiction. These results indicate the importance of mobile phone addiction and the importance of resilience for adolescents. Other studies have also shown that smartphone addiction caused health problems, such as eating, sleeping, and mood disorder [63]. The findings may also help educators and medical personnel distinguish between predictive factors for adolescent mobile phone addiction. They could also be used to design an intervention to effectively treat and prevent mobile phone addiction in adolescents when dealing with future difficult and traumatic events. At the same time, it also indicates the importance of cultivating the psychological resilience of adolescents in coping with pressure and difficulties since having good psychological resilience can reduce depression, anxiety, and reduce mobile phone addiction.

## Supporting information

**S1 Data.**
(XLSX)

## Author Contributions

**Conceptualization:** Hongjuan Chang.

**Investigation:** Xue Li, Shenhua Zhang.

**Writing – original draft:** Anna Ma.

**Writing – review & editing:** Anna Ma, Yan Yang, Shuangxi Guo.

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
