## [Decision Letter · Decision Letter 0]

9 Aug 2022

PONE-D-22-09528

Adolescent Resilience and Mobile Phone Addiction in Henan province of China: Impacts of Chain Mediating, Coping Style

PLOS ONE

Dear Dr. Ma,

Thank you for submitting your manuscript to PLOS ONE. After careful consideration, we feel that it has merit but does not fully meet PLOS ONE’s publication criteria as it currently stands. Therefore, we invite you to submit a revised version of the manuscript that addresses the points raised during the review process.

Your article has been reviewed by two peer-reviewers. Their comments suggests that your manuscript can be strengthened by improvements on the reporting of multiple aspects of this manuscript, such as further clarifications around the use of the term "addiction", better explanation of the methodology, and more in-depth discussion of the limitation of the study.

Could you please revise the manuscript to carefully address the concerns raised?

We look forward to receiving your revised manuscript.

Kind regards,

Maria Elisabeth Johanna Zalm, Ph.D

Editorial Office

PLOS ONE

https://journals.plos.org/plosone/s/file?id=ba62/PLOSOne_formatting_sample_title_authors_affiliations.pdf".

a) Did participants provide their written or verbal informed consent to participate in this study?

“This work was supported by the National Natural Science Foundation of China(grant number: 81803252).”

6. We note that you have indicated that data from this study are available upon request. PLOS only allows data to be available upon request if there are legal or ethical restrictions on sharing data publicly. For more information on unacceptable data access restrictions, please see http://journals.plos.org/plosone/s/data-availability#loc-unacceptable-data-access-restrictions.

7. Please amend the manuscript submission data (via Edit Submission) to include author **Yan Yang, Shuangxi Guo, Xue Li, Shenhua Zhang**

8. Your ethics statement should only appear in the Methods section of your manuscript. If your ethics statement is written in any section besides the Methods, please move it to the Methods section and delete it from any other section. Please ensure that your ethics statement is included in your manuscript, as the ethics statement entered into the online submission form will not be published alongside your manuscript.

9. Please upload a new copy of Figures 1 and 2 as the file does not open. Please follow the link for more information: "" ext-link-type="uri" xlink:type="simple">https://blogs.plos.org/plos/2019/06/looking-good-tips-for-creating-your-plos-figures-graphics/""
https://blogs.plos.org/plos/2019/06/looking-good-tips-for-creating-your-plos-figures-graphics/"".

Reviewers' comments:

Reviewer's Responses to Questions

**Comments to the Author**

1. Is the manuscript technically sound, and do the data support the conclusions?

Reviewer #1: Partly

Reviewer #2: Yes

2. Has the statistical analysis been performed appropriately and rigorously? 

Reviewer #1: Yes

Reviewer #2: I Don't Know

3. Have the authors made all data underlying the findings in their manuscript fully available?

Reviewer #1: Yes

Reviewer #2: Yes

4. Is the manuscript presented in an intelligible fashion and written in standard English?

Reviewer #1: Yes

Reviewer #2: Yes

5. Review Comments to the Author

Reviewer #1: This article may be corrected as follows :

1. In abstract to be improved to describe succinctly related to Introduction, Method, Result and Discussion, on a good abstract will also be seen what becomes novelty of the article.

2. The introduction must be explained two main aspects; there are aspects of the theory that is backgrounded and problems found in the field to look research gap from this article. Then at the end of the introduction will be able to see what is new from this study.

3. The methodology may need to be explained in more detail related to the research subject so that it becomes clear who the subject is.

4. The result and discussion should give the table and figure of results more in-depth, and you have not done.

5. Conclusions must undoubtedly be able to answer from the purpose of this study.

*** I found the strengths of this article, but there are still weaknesses that need to be fixed.

This article is acceptable in PLOS ONE with Major Revision

Reviewer #2: Link between resilience and mobile phone addiction not made clear.

Resilience not well defined, how is it being applied and to what purpose?

What definition of mobile phone addiction? When does use become addiction? Why is this important?

AS well as consent from parents – did young people themselves get a chance to consent or at least give assent?

Last sentence page 3 seems out of place – reads like findings.

Greater focus on the impact of and recovery from crisis such as covid, on mobile phone use, resilience and mental and social, wellbeing would make a clearer rationale for the study.

The impact of covid could have been considered beyond the stress and anxiety it caused. It is natural to expect through lockdowns and social restrictions that adolescents would have become more reliant on mobile phones. The mention of covid here feels somewhat tokenistic and not very well thought through.

Greater explanation of methodology and methods would make it clearer why and how this research was done the way it was.

Discussion of the findings limits the usefulness of the research. It could be made much ore clear what the links are between main components of the research and the future implications of this for Chinese adolescents.

Some written English and referencing issues to be corrected.

6. PLOS authors have the option to publish the peer review history of their article (what does this mean?). If published, this will include your full peer review and any attached files.

Reviewer #1: No

Reviewer #2: No

---

## [Author Response · Author response to Decision Letter 0]

28 Sep 2022

Reviewer #1: This article may be corrected as follows :

1. In abstract to be improved to describe succinctly related to Introduction, Method, Result and Discussion, on a good abstract will also be seen what becomes novelty of the article.

2. The introduction must be explained two main aspects; there are aspects of the theory that is backgrounded and problems found in the field to look research gap from this article. Then at the end of the introduction will be able to see what is new from this study.

3. The methodology may need to be explained in more detail related to the research subject so that it becomes clear who the subject is.

4. The result and discussion should give the table and figure of results more in-depth, and you have not done.

5. Conclusions must undoubtedly be able to answer from the purpose of this study.

Response

Thank you for giving this suggestion.We revise the abstract section and introduction,and detail explain the subject,amend the result,discussion and conclusion.

Reviewer #2: Link between resilience and mobile phone addiction not made clear.

Resilience not well defined, how is it being applied and to what purpose?

What definition of mobile phone addiction? When does use become addiction? Why is this important?

AS well as consent from parents – did young people themselves get a chance to consent or at least give assent?

Last sentence page 3 seems out of place – reads like findings.

Greater focus on the impact of and recovery from crisis such as covid, on mobile phone use, resilience and mental and social, wellbeing would make a clearer rationale for the study.

The impact of covid could have been considered beyond the stress and anxiety it caused. It is natural to expect through lockdowns and social restrictions that adolescents would have become more reliant on mobile phones. The mention of covid here feels somewhat tokenistic and not very well thought through.

Greater explanation of methodology and methods would make it clearer why and how this research was done the way it was.

Discussion of the findings limits the usefulness of the research. It could be made much ore clear what the links are between main components of the research and the future implications of this for Chinese adolescents.

Some written English and referencing issues to be corrected.

Response

Thank you for giving this comments. We have remend the introduction of this article to emphasize definition of Resilience and mobile phone addiction,and explains the application of resilience in many studies.We explain the harm of mobile phone addiction to adolescents.The study complied with ethical requirements and was carried out on the basis of the consent of the parents or guardians of minors and adolescent themselves.We also revise the methods ,result ,discussion and conclusion section, the written English and referencing are corrected in the article.

---

## [Decision Letter · Decision Letter 1]

14 Nov 2022

Adolescent Resilience and Mobile Phone Addiction in Henan province of China: Impacts of Chain Mediating, Coping Style

PONE-D-22-09528R1

Dear Dr. Ma,

We’re pleased to inform you that your manuscript has been judged scientifically suitable for publication and will be formally accepted for publication once it meets all outstanding technical requirements.

Kind regards,

Fahad Jibran, Ph.D.

Academic Editor

PLOS ONE

Additional Editor Comments (optional):

Reviewers' comments:

Reviewer's Responses to Questions

**Comments to the Author**

1. If the authors have adequately addressed your comments raised in a previous round of review and you feel that this manuscript is now acceptable for publication, you may indicate that here to bypass the “Comments to the Author” section, enter your conflict of interest statement in the “Confidential to Editor” section, and submit your "Accept" recommendation.

Reviewer #1: All comments have been addressed

2. Is the manuscript technically sound, and do the data support the conclusions?

Reviewer #1: Yes

3. Has the statistical analysis been performed appropriately and rigorously? 

Reviewer #1: Yes

4. Have the authors made all data underlying the findings in their manuscript fully available?

Reviewer #1: Yes

5. Is the manuscript presented in an intelligible fashion and written in standard English?

Reviewer #1: Yes

6. Review Comments to the Author

Reviewer #1: This article may be corrected as follows :

I concluded that what has been corrected for this article is:

This article has been corrected as a correction from the reviewer,

and after I checked again, it can be concluded that

this article has fulfilled what was requested so that

this article can be accepted and published in PLOS ONE Journal.

But I ask that the author can add some references

that have a similarity to this research, and I hope to shrink some of

the articles that have been published in PLOS ONE for the last three years.

This article is Acceptable in PLOS ONE Journal.

7. PLOS authors have the option to publish the peer review history of their article (what does this mean?). If published, this will include your full peer review and any attached files.

Reviewer #1: **Yes: **Muhammad Ali Equatora

---

## [Editor Report · Acceptance letter]

14 Dec 2022

PONE-D-22-09528R1 

Adolescent Resilience and Mobile Phone Addiction in Henan Province of China: Impacts of Chain Mediating, Coping Style 

Dear Dr. Ma:

I'm pleased to inform you that your manuscript has been deemed suitable for publication in PLOS ONE. Congratulations! Your manuscript is now with our production department. 

Kind regards, 

on behalf of

Dr. Fahad Jibran 

Academic Editor

PLOS ONE